# Cattle Make the Difference: Variations and Developments of Animal Husbandry in the Central European La Tène Culture

**DOI:** 10.3390/ani13111847

**Published:** 2023-06-01

**Authors:** Konstantina Saliari, Peter Trebsche

**Affiliations:** 1Natural History Museum Vienna, 1. Zoological Department, Archaeozoological Collection, Burgring 7, 1010 Vienna, Austria; 2Department of Archaeology, Universität Innsbruck, Innrain 52A, 6020 Innsbruck, Austria; peter.trebsche@uibk.ac.at

**Keywords:** Late Iron Age, La Tène Culture, archaeozoology, centralization, agricultural intensification, cattle husbandry, morphometric analysis

## Abstract

**Simple Summary:**

The analysis of animal bones and teeth from archaeological excavations is of great significance to better understand aspects of past human societies. The present study focuses on the analysis of the Middle La Tène period faunal material from the settlement of Haselbach in Lower Austria, one of the biggest archaeozoological assemblages from present-day Austria. The study of the faunal material exhibits features of urbanization, similar to the settlement of Roseldorf (Lower Austria), 35 km northwest of Haselbach. The archaeozoological results from Haselbach are later compared with other sites from the La Tène period located in Central Europe. The overview of the archaeozoological data suggests major changes especially during the Middle La Tène period probably related to agricultural intensification. Furthermore, after studying the biological profiles (age and sex profiles) of all major domesticated species, especially the age and sex distribution of cattle were used to distinguish different patterns of cattle husbandry. Finally, the presence of different animal populations (especially in the case of cattle) offers crucial evidence on long-distance animal exchange and the growing influence of the South during the pre-Roman late Iron Age.

**Abstract:**

The first part of our research focuses on the analysis of animal remains (>6000 identified specimens, NISP) from the Middle La Tène central settlement Haselbach in Lower Austria, one of the largest investigated archaeozoological assemblages of present-day Austria. Based on the age and sex profiles, the faunal assemblage from Haselbach shows characteristics of urbanization and centralization and bears striking similarities to the archaeozoological material of the central settlement of Roseldorf (Lower Austria), some 35 km northwest of Haselbach. The second part of our research discusses the historical and regional context of the archaeozoological results from Haselbach and compares them with other sites, based on a detailed review of published archaeozoological data from the La Tène period (c. 450 BC to the end of the first century BC). In total, 55 faunal assemblages from 46 sites in nine countries in Central Europe, representing different types of sites (lowland settlements, hilltop settlements, central settlements, oppida, assemblages of ritual activity, and mining sites) were examined. The synthesis of the archaeozoological data exhibits different husbandry strategies and suggests major changes, especially during the Middle La Tène period indicating agricultural intensification. The differences in the biological profiles of the major domesticated species are of crucial importance to better understand aspects of socio-economic organization; especially in the case of cattle, age and sex profiles are used to distinguish different patterns of cattle husbandry. Finally, morphometric and recent genetic analyses on cattle bones and teeth from La Tène sites in Central Europe provide new insights into the complex socio-economic behavior as well as long-distance networks, involving animal supply and mobility in an exciting period of change involving centralization and increasing influence from the South during the pre-Roman late Iron Age.

## 1. Introduction

From the 5th to the 1st centuries BC, the La Tène Culture experienced fundamental changes in settlement structure. For a long time, the oppida of the late 2nd and 1st century BC were regarded as the heyday of this development (“the earliest towns north of the Alps” [1]). However, archaeological research of the past three decades showed that agglomeration processes of lowland settlements began much earlier, namely in the 3rd century. Until the 1980s, this phenomenon was mainly conceptualized as a process of urbanization, later as proto-urbanization, centralization, or nucleation process (e.g., [2,3,4,5,6]). Within these frameworks, the role of artisanal production, minting of coins, and long-distance trade has been strongly emphasized, while the agricultural basis has been rather neglected. The study of animal bones from settlement contexts substantially contributes to characterizing the mentioned processes, as an indicator of animal husbandry regimes and as evidence of meat consumption or provision.

In our study, we investigate both aspects presenting a new unpublished case study and summarizing existing archaeozoological data. The first part consists of an archaeozoological analysis of the Middle La Tène period settlement center at Haselbach in Lower Austria. The second part provides an intensive literature survey of existing archaeozoological data from the Eastern La Tène Culture that has never been synthesized before on this scale. Our approach is to confront archaeozoological data with settlement categories and different find contexts (such as ritual or mining sites), along the timeline of the La Tène Culture.

The area of study is the Eastern La Tène Culture that extended over eastern Austria, Bohemia, and Moravia in the Czech Republic, southern Poland, Slovakia, western Hungary, western Romania, Slovenia, Croatia, and northern Serbia (Figure 1). The settlement system of this region can be briefly characterized as follows [7,8,9,10,11,12,13]: In the Eastern La Tène Culture, small unfortified lowland settlements were almost the only known category during the Early La Tène period (phases LT A and B, ca. 450–250 BC), alongside with very few known hilltop settlements. At the beginning of the Middle La Tène period (phase LT C, ca. 250–150 BC), large settlement agglomerations emerged [14,15]. The largest settlement centers typically extended over some dozens of hectares and functioned as centers of cult and ritual, as mints for the production of gold and silver coinage, and/or as centers of artisanal production and distribution [16]. In the so-called Amber corridor region (i.e., southern Poland, Moravia, and Eastern Austria), which forms one of the core areas of development in the Eastern La Tène Culture, a clear hierarchy of settlements according to their size and number of sanctuaries established. The settlement system comprised large settlement centers (ca. 40 hectares, e.g., Roseldorf AT, Němčice CZ or Nowa Cerekwia PL) and medium-sized centers (ca. 6–10 hectares, e.g., Haselbach AT), followed by common villages (1–3 hectares) and dispersed single farmsteads [10,11]. After this period of population growth and increasing settlement density in the second half of the 3rd century and the first half of the 2nd century, most of these agglomerations located in very fertile regions were abandoned. Instead, hillforts of different sizes (the larger ones usually labeled oppida) were established in the Late La Tène period (phase LT D, ca. 150 BC until the time of Roman occupation). Even larger lowland settlements (up to 1 km^2^) grew mainly at important river crossings, while smaller villages and farmsteads continued to exist.

Was settlement growth the cause or the consequence of changes in agricultural production? According to the prevailing paradigm in economic archaeology, developed by Esther Boserup in her seminal book on “The conditions of agricultural growth” (1965), farmers will counteract population growth and increasing population density by becoming more productive in one way or another—i.e., through intensification of agricultural production which is regarded as the precondition of the transition from a subsistence economy to an urban economy [17]. Following a functional definition of urbanism, i.e., asking “what a city does” instead of “what a city is”, we aim at investigating the links between settlement types, agricultural regimes, and different meat consumption patterns (cf. [18]). Were changing settlement structures accompanied by changing husbandry regimes? Did meat consumption differ according to settlement category or functional areas (like cult sites)? Did inhabitants of small settlements consume different meat compared with those from large settlement centers? The answers will provide a better understanding of nucleation and urbanization processes. As mentioned above, we will start with a close look at a case study of a medium-sized settlement center of the Middle La Tène period.

## 2. The Case Study of Haselbach (Lower Austria)

### Materials and Methods

The case study focuses on the site of Haselbach “Im äußeren Urban”, which belongs to the municipality of Niederhollabrunn, district of Korneuburg in the state of Lower Austria (Figure 2; geographical coordinates: E 16,24415 N 48,45212, altitude 212 m above sea level). Located in the middle of fertile loess soils, about 11 km north of the river Danube, the site occupies an area of approximately 6.6 hectares according to geomagnetic surveys [11,19]. The surveys revealed approximately 119 rectangular sunken features (so-called pit dwellings or semi-sunken buildings), ca. 109 pits (probably storage pits), one square enclosure, and about 31 irregular features. Based on the geophysical prospections, a French-Austrian team of archaeologists under the direction of Peter Trebsche and Stephan Fichtl conducted targeted excavations in five areas (ca. 10%) of the settlement from 2015 to 2019 [20,21,22,23,24]. The excavation aimed at identifying the function of the sunken buildings. Their preserved floors were systematically sampled for micromorphological and geochemical analyses. All prehistoric fill layers were sampled for the retrieval of archaeobotanical macro-remains and other micro-finds by flotation and wet-sieving. According to the preliminary analyses of pottery, small finds (glass artifacts, brooches), coins, and a series of 36 radiocarbon dates, the settlement was founded at the beginning of phase LT C and ended at the very beginning of LT D1.

The archaeozoological analysis investigates the faunal remains from area 1 and area 2 at the site of Haselbach, found during the excavation campaigns 2015–2016, which represent approximately one half of the total weight of animal bones recovered in all five campaigns. The late Iron Age features from area 1 yielded 2107 identified animal bones and teeth with a total weight of around 36 kg, whereas the material from area 2 consists of 4074 identified animal bones and teeth weighing almost 74 kg. Although the faunal assemblage was retrieved at a relatively good state of preservation, in a few cases the surface of the animal bones was slightly eroded or exhibited gnawing marks and/or recent damage that took place during the excavation. The existence of butchery marks and fractures on most of the bones suggests that they mainly represent food waste.

The identification of the faunal remains was carried out at the Natural History Museum Vienna (1. Zoological Department, Archaeological-Zoological Collection) using the Osteological Collection and the Adametz Collection. Osteological criteria were used to separate the bones of sheep/goats [25,26,27,28,29,30,31,32,33,34,35,36].

The age estimation was based on the epiphyseal fusion of the bones and on the eruption and wear stages of the maxillary and mandibular deciduous premolar Pd4 and permanent molar M3 [37]. These teeth were chosen because their identification is relatively easy, and they share only a small period of coexistence. The system followed for noting the wear stage was based on four different stages: 0 (no wear), + (slightly), ++ (medium), and +++ (significantly).

Sex estimation for ruminants was addressed based on horn cores and pelves [25,38,39,40]. For cattle, the metapodials were additionally studied [39,41,42,43,44,45]. Canini teeth and tooth sockets (alveoli) were used for sexing pigs [46,47]. The skeletal element representation was based on the number of identified fragments (NISP) for better compatibility with other sites.

The animal bones were also morphologically studied and compared with other faunal assemblages. Whenever it was possible, the measurements that were taken according to the standard of von den Driesch (1976) were statistically processed [48]. The size of the animals was calculated according to the height at withers, following the factors set for each species [49,50,51,52,53].

Modifications including butchery marks and gnawing marks were recorded. Concerning butchery marks, their type, orientation, and location were documented. Their interpretation was based on several published works [54,55,56,57,58,59,60,61].

## 3. Results

### 3.1. Species Representation

In total, 2107 identified faunal remains were found in area 1 and 4074 in area 2. Based on NISP data, domesticated taxa prevail in area 1 with 2105 animal bones and teeth (Table 1, Figure 3). Only two fragments could be attributed to wild species, one to the European hare (*Lepus europaeus*) and one to the European pond turtle (*Emys orbicularis*). Among the domesticated species, sheep/goat prevailed (39.1%), followed by pig (36.8%), cattle (18.9%), horse (2.9%), and dog (2.2%). Similar results were obtained from area 2 (Table 2 and Figure 3). In area 2, most faunal remains derive from domesticated taxa with 4071 animal bones; only three bones were attributed to wild species: roe deer (Capreolus capreolus), red fox (Vulpes vulpes), and fish. In area 2, sheep/goat dominated (45.9%) followed by pig (34.7%), cattle (14.9%), dog (2.6%), and horse (1.8%). Concerning the distribution of sheep/goat, sheep prevailed in both areas with 79.0% (area 1, NISP: 163) and 78.5%, (area 2, NISP: 362), respectively.

Based on weight analysis (Figure 3) from area 1, cattle was the dominant species (36.3%), followed by pig (30.3%), sheep/goat (23.5%), horse (8.5%), and dog (1.4%). Similarly, in area 2 cattle was by weight the most important species (33.7%) followed by pig (32.0%), sheep/goat (26.3%), horse (6.4%), and dog (1.6%). The combination of the data from both areas shows the dominance of sheep/goats according to NISP (43.5%), but the prevalence of cattle (34.6%) by weight (Figure 3).

### 3.2. Skeletal Element Representation

The skeletal element representation for the three economically important species (cattle, sheep/goat, and pig) is inferred by Table 1 and Table 2. Although no significant discrepancies were noted, some variations were observed in the case of sheep/goats and pigs. In particular, concerning sheep/goat a high number of ribs (26.3%) in combination with a very low percentage of vertebrae (2.7%) was noted; in the case of pig, a very low number of ribs (6.7%) was documented. These differences in the abundance of ribs and vertebrae might be attributed to taphonomical processes or the distribution of different parts of the animals—for example—in different parts of the settlement which have not been excavated yet.

### 3.3. Age and Sex Distribution

Based on the epiphyses, most cattle and sheep/goat individuals derive from adult animals with 83.5% and 89.9%, respectively. Adult individuals prevailed for pigs as well (61.5%), but the number of non-adult pigs was slightly higher (38.5%). More information could be obtained by the study of teeth. Concerning cattle, seven teeth were recorded (Figure 4): two teeth suggest individuals between 1 and 2 years (Pd4++), one tooth points to an animal between 5 and 7 years (M3+), and four teeth between 7 and 10 years (M3++). Additionally, one cattle cranium and one cattle maxilla suggest two individuals between 7 and 10 years (M3++). Significantly more teeth were recorded for sheep/goats (n: 152) and pigs (n: 127). The age profile for sheep/goats shows that they were represented by all age stages and that adult individuals prevailed (Figure 4). Concerning adults, two different peaks could be observed: a high number of animals were slaughtered between 3 and 5 years (M30, 28.3%) and 7–10 years (M3++, 28.3%). Pigs were also represented by all age stages, but young adults prevailed (M30, 35.4%) (Figure 4).

Concerning sex distribution, male/castrated cattle individuals dominated (53.1%); oxen were recorded with 48.4%, males with 4.7% and females with 46.9% (Table 3). The sex distribution for sheep indicates the dominance of females with 53.9% (Table 4). Similar results were documented for goats (females with 54.5%) and for bones only identified as sheep/goat (females with 54%) (Table 4). Based on the alveoli, pigs show a slight prevalence of females (51.9%) (Table 5).

### 3.4. Morphological Observations and Size Reconstruction

The calculation of height at withers for cattle individuals was calculated between 98.3 cm and 113.1 cm for female and castrated animals (n: six metacarpals, one metatarsus), suggesting the presence of the small-sized Iron Age cattle population, also supported by the metric data (see the comparative study, chapter 4.3). However, two cattle bone fragments exhibited significant morphometric differences compared to most cattle remains from Haselbach: one male metacarpus and one metatarsus from a castrated individual (Figure 5) were significantly larger and more robust, bearing striking similarities to the large-sized cattle morphotype, commonly recorded in present-day Austria during the Roman period [62,63,64,65,66,67]. Although the fragmentary state of the bones from Haselbach did not allow exact measurements for these two fragments, the macroscopic appearance shows significant morphological similarities with cattle remains from Roman period sites (Figure 5).

Concerning sheep, the height at withers was reconstructed between 52.2 cm and 68.2 cm with an average of 61.9 cm, based on 41 bones (radius n: 1, metacarpus n: 12, tibia: 2, talus n: 16, metatarsus n: 10). The height at withers for pigs was reconstructed between 65.3 cm and 76.1 cm with an average of 70.2 cm, based on ten tali and at 78.4 cm according to one radius.

The height at withers for dogs could be calculated only for one individual (tibia) at 47.2 cm. However, metric data suggests the existence of larger dogs, too (see comparative study, chapter 4.3). Finally, the height at withers for horses between 113.2 cm and 130.7 cm (metacarpus n: 1, metatarsus n: 1) suggests the existence of the small-sized La Tène horse.

### 3.5. Modifications: Butchery Marks

The animal bones from Haselbach exhibited a high number of chop marks, indicating disarticulation, dismemberment, portioning, and marrow extraction. Only a few cut marks were noted, suggesting skin removal and further dismemberment (Figure 6a–c).

Chops on cattle crania indicate the removal of the horns and the separation of the skull into two parts. Chop marks on the maxillae, zygomatica, occipital condyles and the mandibles (caput mandibulae, processus coronoideus, tuberositas musculi sternomandibularis and corpus mandibulae) suggested further separation. A plethora of chop marks was documented on the ribs and the vertebrae. Marks on the vertebrae mainly indicate the removal of the articulation surfaces or of the processus spinosus and transversus on the lumbar and thoracic vertebrae; diagonal and longitudinal chop marks were also documented on the cervical vertebrae. Diagonal and transversal chops were found on many areas of the pelves including ilium, acetabulum, symphysis, and foramen obturatum. Longitudinal chops were documented on the cavitas glenoidalis and spina scapulae of scapulae, whereas diagonal marks were noted on the processus coronoideus, fossa supraspinata, and fossa infraspinata. Long bones were chopped at different parts on the midshaft and sometimes on the joints. Longitudinal chop marks on humeri, radii, femora, tibiae, and metapodials suggest marrow extraction. Cuts on metatarsals (proximal part), tali, and first phalanges point towards further dismemberment. Cuts on the midshaft of one metacarpus revealed skin removal (Figure 6a).

Concerning sheep/goat, chops on the cranium show the removal of horns and the splitting of the skull into two parts, similar to cattle. Further chops were detected on the diastema and ramus mandibulae of the mandibles. Ribs were chopped from 5 to 10 cm long parts. Diagonal chops were documented on scapulae and the long bones. Radii, tibiae, and metapodials were longitudinally chopped suggesting marrow extraction. A low number of mandibles and metapodials were found whole (Figure 6b).

Chop marks on pig crania indicate the splitting into two parts, similar to cattle and sheep/goats. Further chops suggest the removal of os incisivum and additional chopping of the frontal bone and the maxillae. Chops on the os zygomaticum were attributed to the acquisition of cheek meat. Chop marks were also found on mandibles and long bones. Diagonal chops were noted on the joints of several long bones, whereas some of them (humerus, femur, tibia) show longitudinal chops, indicating marrow extraction. More rarely the small-sized metapodials were also chopped. The pelves exhibited chop marks on the ilium and the symphysis. Fractures on the fossa supraspinata und fossa infraspinata were usually documented on scapulae. The ribs were chopped into smaller portions, similarly to sheep/goat (Figure 6a–c). Finally, butchery marks were noted on various bones of dog and horse, showing consumption of their meat (Figure 6b,c).

## 4. Review of Archaeozoological Data from La Tène Sites in Central Europe

To create a data basis as sound as possible, we conducted an intensive systematic survey of published archaeozoological data from Austria, the Czech Republic, Slovakia, Hungary, Romania, Poland, Slovenia, Croatia, and Serbia. We included all late Iron Age faunal assemblages from the La Tène Culture, which were retrieved from settlement contexts in the broadest sense (including ritual sites and sanctuaries) and which contained more than 100 identified mammal bones and teeth. Animal bones from burials were not included in this study, because they reflect a selection based on burial customs.

As a result of the literature survey, we present 55 assemblages from 46 sites in nine countries (Table 6). The state of archaeozoological research varies considerably in the area under study, depending on national traditions, the existence of archaeozoological research facilities, and the specialization of archaeozoologists in different epochs. An excellent starting point were recent syntheses of archaeozoological data from Austria [68], Hungary [69], Moravia [70], Table 6, Bohemia [8], Slovakia [71], and Poland [72]. We are also grateful to many colleagues for providing publications and information on the state of research (L. Bartosiewicz, Z. Bielichová).

The geographical distribution of archaeozoologically investigated assemblages is very uneven. The dataset is very unbalanced and certainly not representative of all types of settlements, with hillforts and oppida being underrepresented; therefore, two large oppida from Southern Germany (Manching and Altenburg) were included for comparison.

Collecting, comparing, and interpreting the rich archaeozoological data of this area provided important insights into the common features as well as differences of the La Tène (LT) faunal assemblages. For reasons of comparability and to better understand the variability of husbandry practices of the various species, the reviewed assemblages were studied chronologically and based on their archaeological context on site or even smaller level. Thus, based on the literature cited in the introduction, the sites were separated into (i) lowland settlements (settlements without fortification, <6 hectares or of unknown extent; n: 38 assemblages), (ii) hilltop settlements (based on topography, with or without fortification; n: 4 assemblages), (iii) lowland central settlements (without fortification, >6 hectares, with evidence of central functions, e.g., minting, sanctuaries, artisanal production; n: 2 assemblages), (iv) oppida (protected topography, with fortification, >12 hectares, with evidence of central functions; n: 6 assemblages), (v) ritual contexts/sanctuaries (with depositions of numerous weapons and human bones, mostly within enclosures of different size; n: 4 assemblages) and (vi) mining sites (one of the best-studied site being the salt mine at Dürrnberg in Austria). Table 6 contains an alphabetical list of all studied assemblages with relevant references.

Of course, the publications of faunal material from different sites and regions represent a heterogenous data set, especially because of the highly variable number of bones from different sites and the obvious bias of data deriving from different archaeozoologists, partly using different approaches; all these heterogeneities were kept in mind during data review. For example, the number of identified specimens (NISP) was used for the comparison of the data (e.g., species representation, age, sex distribution, and skeletal element representation), because NISP is commonly used. Concerning age reconstruction, most analyses are based on bone epiphyses and in several cases teeth. The different systems of aging (especially in the case of teeth) could not always be directly compared, thus bigger entities were used (immature, juvenile, subadult, young adult, adult) for better compatibility. Another challenge was related to sex reconstruction, since it was not always clear which elements were used for sexing. For example, in the case of pigs, it was not always clear if sexing was based on teeth (canini) or alveoli, which is important information for archaeozoological interpretation. This is because the existence of lose canini is not always connected with the existence of the animals at the site. Canini might have been collected for the manufacture of artifacts. Morphometric comparative studies could take place only when measurements, the height at withers, or other relevant observations and comments were available.

### 4.1. Species Representation

A collective overview of all sites studied (Table 6) shows that cattle dominated most La Tène assemblages (37 out of 55). However, noteworthy variations were documented depending on the archaeological context, type, and function of site. NISP data show that cattle remains were generally better represented in oppida (n: 6 assemblages), hilltop settlements (n: 4 assemblages), ritual contexts (n: 4 assemblages), and the mining site of Dürrnberg (Figure 7).

The percentage of cattle bones seems to decrease during the Middle La Tène (MLT) period (Figure 8a) in some lowland settlements (n: 38 assemblages), whereas the numbers of sheep/goats and pigs increase in percentage [70,71,81,82,83,85,90,94,96,98,103]. These observations are also valid for the site of Roseldorf (Figure 8b)—a lowland settlement interpreted as a central settlement and thus included in the latest category—where a numerical prevalence of sheep/goats has been noted ([86]). This tendency is further supported by the newly studied material from Haselbach as it has been already shown, where sheep/goats were numerically better represented (Figure 8b).

Horses were mostly represented in assemblages from ritual contexts and in oppida (Figure 7 and Figure 8c–d) [70,86,92,93,100]. A higher number of dog bones (Figure 8a,c) was found at some lowland settlements and oppida [92,93,107]. Domesticated birds were usually represented in very low percentages in all site types (Figure 8a–e).

Although hunting activities were documented at several sites, the percentage of wild taxa is usually low. The highest percentages of wild animals derive from some lowland settlements (Figure 8a): Bořitov (10.8%), Mitterretzbach (both phases with 9.5% and 14.9%) and Sajópetri (15.7%), and hilltop settlements (Figure 8e): Gomolava (6.6%), Liptovská Mara II (11.5%), Nitra-Hrad/Východné nádvorie (9.8%).

### 4.2. Sex and Age Distribution

#### 4.2.1. Cattle

In total, data on sex reconstruction were available from 17 assemblages (Figure 9a). These data show that cattle individuals from lowland settlements were mainly represented by female animals (52.6–80%), except for Michelndorf, where male individuals prevailed [81]. The percentage of male/castrated cattle slightly increased during the MLT period up to approximately 40% (Figure 9a): in Göttlesbrunn male/castrated cattle were present with 39.5% [79], in Mitterretzbach (MLT) with 38.5% [83], in Mšecké Žehrovice (LTC2-D1) with 42.9% [89] and in Nitra-Šindolka with 37.5% [111]. In contrast, in large lowland settlement centers such as Roseldorf ([86]) and Haselbach male/castrated animals dominated (67.6% and 53.1%). Sex distribution for assemblages from ritual contexts (Roseldorf-Great Sanctuary and Frauenberg) shows a prevalence of male/castrated cattle with 83.5% and 83.3% (Figure 9a), respectively [78,86].

Concerning age at slaughter, a higher number of older adult cattle was generally notable. A high percentage of cattle remains in Roseldorf-Great sanctuary (30.7%) and Frauenberg (37%) were represented by animals slaughtered between 7 and 10 years [78,86]. Similar results derive from Haselbach. Only in Mšecké Žehrovice (LT C2-D1) a higher number of younger cattle was noted [89].

Sex reconstruction from the oppida of Manching and Altenburg in Germany shows a dominance of female cattle (Figure 9a), similar to several lowland settlements [99,100]. It is noteworthy however that in Altenburg male/castrated cattle reached a high percentage with 45.3% [99]. In contrast to most lowland settlements, a relatively high number of younger cattle (younger than five years) was documented.

Finally, at the mining site of Dürrnberg female cattle prevailed with 77.9% [73]; a higher number of adults (35.3%) represented older individuals (7–10 years).

#### 4.2.2. Sheep/Goat

Fewer data-sets on age and sex distribution exist for sheep/goat (n: 10 assemblages). Female individuals dominated in almost all archaeological contexts that could be examined (oppida: Manching, Altenburg; lowland settlements: Inzersdorf, Michelndorf, Nitra Šindolka; central settlements: Roseldorf, Haselbach; ritual contexts: Roseldorf-Great Sanctuary, Frauenberg; mining site: Dürrnberg). The sanctuary at Frauenberg, where male/castrated sheep/goats dominated (Figure 9b) constitutes an exception. In contrast to cattle, sheep/goats from several lowland settlements such as Giarmata, Inzersdorf, Michelndorf, and Oberschauersberg were slaughtered more frequently at a younger age stage [80,84]; [81,107]. Similar results have been described from the two ritual sites Frauenberg [78] and Roseldorf-Great Sanctuary [86]. At the site of Dürrnberg adults were mainly represented by individuals between 5 and 7 years. A higher number of older adult animals (older than 7 years) was noted at some lowland settlements such as Nitra Šindolka [111] and Szakály-Réti Földek [105], the central sites of Haselbach and Roseldorf-settlement [85], the oppidum of Manching [100].

#### 4.2.3. Pig

Data on sex distribution for pigs derive from 14 assemblages (Figure 9c). Sex distribution for pigs varies significantly among the different assemblages. At lowland settlements both males/castrated and females may prevail (Figure 9c). At the central settlements of Roseldorf [85], Haselbach, and the oppidum of Manching [100] the percentage between male/castrated pigs and females was almost equal, whereas at the oppidum of Altenburg [99] and the mining site of Dürrnberg females dominated (57% and 64.3%). Differences were also noted in ritual contexts (Figure 9c): in Roseldorf-Great Sanctuary [86] females prevailed (60%), whereas at the sanctuary of Frauenberg [78], males/castrated reached very high numbers (87.2%). Pigs were usually slaughtered between 1.5 and 3 years [73,78,81,82,85,90,99,100,105,107,108].

### 4.3. Morphometric Analysis

#### 4.3.1. Cattle

La Tène cattle remains generally show the existence of small-sized individuals as noted in Dunaszentgyörgy [101], Göttlesbrunn [79], Inzersdorf [80], Liptovská Mara [109]; [71], Michelstetten [82], Nitra Šindolka [111], Radovesice [90], Roseldorf [85,86], Sajópetri [102], Szakály-Réti Földek [105] and Wangheim [87] with a height at withers between 95 and 125 cm (for bulls, oxen and cows). A size reduction was noted at Radovesice from an average of 108.5 to 104.9 cm during the La Tène period [90].

Despite this relative morphometric homogeneity of La Tène cattle bones, fragments of large-sized and more robust cattle were discovered in some Iron Age assemblages, including Manching [100], Wien-“Palais Rasumofsky” [113], Roseldorf [85,86] and Szakály-Réti Földek [105]. Recent genetic analyses on finds of large-sized cattle from Roseldorf-Great Sanctuary and Palais Rasumofsky confirmed their genetic difference to the late Iron Age small-sized cattle, suggesting cattle mobility and imports from the Mediterranean already during the Middle La Tène period ([114]). A very similar large-sized cattle morphotype was widely and systematically present during the subsequent Roman period [62,63,64,66,67,115].

The morphometric results and the size reconstruction from Haselbach fit very well with the archaeozoological context of the La Tène period (Table 7 and Table 8). Additionally, the comparison of various measurements from different cattle remains indicates that cattle from Haselbach exhibited a range of variation very similar to other La Tène period sites (Table 8), suggesting the presence of small-sized Iron Age cattle. Remarkable similarities are noted especially to Roseldorf and Wien-“Palais Rasumofsky” due to the possible presence of “exotic cattle” from the South.

Finally, a common observation for the small-sized Iron Age cattle population concerns the lower third molar: a reduction of the talonid of the third mandibular molar was noted in several sites, including Altenburg [99], Dürrnberg [73], Mšecké Žehrovice [89], Manching [100] and Radovesice [90]. Taking into consideration that teeth are very conservative from an evolutionary perspective [116], this trait might provide some additional information about the origins and distribution of the Iron Age cattle morphotype(s).

#### 4.3.2. Sheep/Goat

Concerning sheep (which usually constitute the majority of small ruminants) the assemblages of Giarmata [107], Göttlesbrunn [79], Inzersdorf [80], Michelstetten [82], Mšecké Žehrovice [89], Oberschauersberg [84], Radovesice [90], Sajópetri [102], Szakály-Réti Földek [105] and Wangheim [87] suggest animals with a height at withers between 52 and 68.0 cm. Similar results derive from Haselbach (Table 9); a comparison of the height at withers (Table 9) and the metric data (Table 10) with other La Tène assemblages (especially in eastern Austria) shows similar values.

Sheep remains from Mitterretzbach [83] and Dürrnberg [73,74,75,76,77] showed significantly larger sheep (77 cm), a size usually present during the Roman period (Table 10); the interpretation of these rare finds of large-sized sheep during the La Tène remains enigmatic [117,118].

#### 4.3.3. Pig

Height at withers of pigs were documented from Dunaszentgyörgy [101], Inzersdorf [80], Michelstetten [82], Mitterretzbach [83], Nitra Šindolka [111], Oberschauersberg [84] and Sajópetri [102] and show values mainly between 69 and 83 cm. In Sajópetri, some larger individuals (93.9 cm) were also documented [102]. Concerning pig remains from Haselbach, the comparison with other La Tène sites shows that the average size from Haselbach is comparatively low (Table 11); however, pigs are plastic animals and their morphology might easily be affected by local factors such as ecology and fodder [119]; [120]. Additionally, bones such as tali and metapodials are not ideal bones for size reconstruction in the case of pigs. A comparison of other metric data (Table 12) shows that the pig remains from Haselbach do not differ significantly from those from other La Tène period sites.

#### 4.3.4. Dog

La Tène period dog bones suggest the presence of several morphotypes with significant size variations: for instance, in Manching [100] the height at withers was reconstructed between 30 and 65 cm. Middle to large-sized individuals, between 46 and 61.5 cm were found in Inzersdorf [80], Michelndorf [81], Michelstetten [82], Nitra Šindolka [111], and Radovesice [90]. A similar wide range of variation was noted in Haselbach. Although the height at withers could be reconstructed only for one individual at 47.2 cm. The review of the metric data shows the existence of different morphotypes: the length of the cheektooth row (M3–P1) for instance suggests the existence of some large-sized dogs (Table 13). In some cases, remains show morphological similarities to what is known today as the German shepherd dog [73]. Exceptional dog finds include the very small-sized dog remains from Roseldorf, with distinct similarities to the Roman period lapdogs [86].

#### 4.3.5. Horse and Ass

Concerning horse, the La Tène period is widely known for small-sized horses as documented in Bořitov [94], Göttlesbrunn [79], Inzersdorf [80], Kobarid-Bizjakova hiša [109], Michelstetten [82], Mitterretzbach [83], Nitra Šindolka [111], Sajópetri [102], Závist [93]; [92], and Haselbach (present study) with a height at withers mostly between 120 and 133 cm. Higher values were noted in Nitra Šindolka with 136.4 cm [111], Radovesice with 141.3 cm [90], and Szakály-Réti Földek with 139.3 cm [105]. Additionally, the find of a domesticated ass at the lowland settlement of Szakály-Réti Földek has been interpreted as an influence from the South [105].

## 5. Discussion

### 5.1. The Case Study of Haselbach

#### Husbandry and Exploitation of the Major Economic Domesticated Species (Cattle, Sheep/Goat, Pig) in Haselbach

Concerning cattle, the high number of castrated individuals from Haselbach is quite remarkable. The normal birth rate of cattle is approximately 50% females and 50% males. The study of archaeozoological assemblages shows that the distribution of different sex is at a later stage influenced by the peasants, related to their specific husbandry organization [121]. For example, faunal assemblages from Iron Age sites in Austria show that, in an autarchic peasant economy of that time, female animals reach a higher percentage than males/castrated (e.g., [79]). Thus, the higher numbers of males/castrated (53.1%) in Haselbach indicates that at least some individuals have been delivered from outside.

The presence of all cattle elements at the site indicates that cattle were represented (and some have been delivered) by whole individuals. Additionally, the age profile suggests a higher number of adults: cattle teeth (n: 7) together with one cattle cranium and a maxilla fragment [122] indicate a higher percentage of older animals between 7 and 10 years (n: 6). This age stage suggests secondary exploitation of most cattle individuals. This is further supported by the pathological alteration observed on a horn core from an ox, which exhibits depressions at the ventral side, indicating its role as a working animal. These observations suggest that older animals were slaughtered after secondary exploitation as working animals or for other products, such as milk in the case of female cattle.

The few indications of an “exotic” large-sized cattle morphotype are important, because the number of documented examples from the La Tène period is still very low. Their origin has been sought in the South (e.g., [115]), which has been confirmed by genetic data recently [114]. As archaeozoological data from Italy show, an increase in cattle body size occurred before the Roman conquest, already during the Iron Age [123]. The presence of this cattle morphotype suggests La Tène animal mobility, and exchange networks and it remains an exciting topic for more detailed studies in Haselbach and other sites.

The most important biological features of sheep/goat in Haselbach are the dominance of females, the wide age profile, and the prevalence of adult animals, which are in contrast with those of cattle. The age profile shows two peaks of age groups: (a) animals between 3 and 5 years and (b) between 7 and 10 years. The first age stage of young adults could be connected with meat consumption of good quality, whereas older individuals suggest secondary exploitation. In total, a slight emphasis on meat is observable, since the total number of animals slaughtered between 3 and 5 years is higher (68.5%). Concerning secondary exploitation, in the case of sheep which are significantly better represented than goats, the high frequency of male/castrated individuals points towards wool exploitation [86,117].

Finally, the wide age spectrum of pigs and the almost equal distribution of males and females in Haselbach is regarded as typical for peasant economies [79,80,82]. In contrast, the selection and clear preference for young animals (54.3% were slaughtered between 1½ and 2 years) together with the very low number of older individuals suggests very good meat quality and constitutes a common characteristic for sites of urban character [85].

##### Haselbach in the Broader Archaeozoological Context of La Tène Period Central Europe

A comparison of the archaeozoological results from Haselbach with other La Tène sites shows striking similarities with the central settlement of Roseldorf [85], which is 35 km towards the northwest. Some of the most important common characteristics are: (a) the numerical dominance of sheep/goat, and especially sheep, (b) the higher number of females among sheep/goat and the slight emphasis on their use for meat, (c) focus on wool exploitation, (d) the importance of cattle for meat supply based on the weight analysis, (e) the dominance of oxen with older individuals, (f) the presence of large-sized (imported) cattle together with the small-sized Iron Age cattle morphotype, (g) the slaughter of pigs at the optimal age stage for meat production. The only difference observed between the two sites concerns the sex distribution of pigs; females prevail in Roseldorf-settlement, whereas in Haselbach females and male/castrated animals are almost equally distributed. Based on the archaeozoological interpretation both sites exhibit a mixture of urban and rural characteristics.

Roseldorf-settlement and Haselbach are both located in northeastern Lower Austria at a similar altitude and show the same tendency observed for lowland settlements, where sheep/goats and pigs are better represented since the Middle La Tène period. The numerical prevalence of sheep/goats at some lowland sites has sometimes been viewed as an indicator of social inequalities and/or economic difficulties [85]. Reconstructing socio-economic structures and internal hierarchies for past societies is still very challenging. However, the study of the faunal remains from Haselbach and Roseldorf-settlement does not imply economic difficulties; on the contrary, the existence of “exotic cattle” from the South does not indicate economic hardship [114]. According to research about the agrosystems of the 19th century in Lower Austria [124,125], it seems that the keeping of sheep in the areas of Roseldorf and Haselbach is favored by the relatively flat and easily accessible landscape as well as land utilization [124,125]. In Lower Austria, cattle breeding would have been more favorable in the northern Waldviertel and the Alpine region [124,125].

### 5.2. Review of Archaeozoological Data from La Tène Sites in Central Europe

#### A Synthesis of the Archaeozoological Observations: Self-Sufficiency, Supply, Agricultural Intensification, and Animal Mobility

Despite the challenges when comparing and interpreting archaeozoological data from numerous archaeozoological assemblages, with different ecological, climatic, and topographic setting, availability of natural resources as well as different local cultures and traditions, our comparison shows that some common tendencies can be traced, although faunal remains from similar types of sites may show considerable variations. For example, the age and sex profiles of the three economically most important species, cattle, sheep/goat, and pig, show that the different taxa were managed differently, and that animal husbandry was dependent on various factors. 

The combination of sex and mortality profiles constitutes an essential tool to interpret economic strategies and to understand the logistical organization of a site. Although all types of contexts were discussed in the text, hilltop settlements are excluded from this section, due to a lack of sufficient material for reconstructing and comparing age and sex profiles.

Lowland settlements and central settlements

Concerning lowland settlements, female individuals were usually more common than males and castrated cattle (e.g., Göttlesbrunn, Inzersdorf, Michelstetten, Mitterretzbach, Sajópetri). Based on the combination of age and sex profiles, some lowland settlements exhibit features that are more characteristic of a rural economy (e.g., Göttlesbrunn, Inzersdorf, Michelstetten, Mitterretzbach), including the prevalence of females, a wide spectrum of age stages and a higher number of adult animals. However, the situation seems to be more complex, because the economic structure among the lowland settlements might differ significantly, as will be shown below.

A key change in the biological profiles of cattle can be observed at some lowland settlements (Mšecké Žehrovice, Göttlesbrunn, Michelndorf, Michelstetten, Mitterretzbach), where the percentage of male/castrated animals increased during the Middle La Tène period. A detailed examination of male cattle bones (horn cores, pelves, metapodials) from that period shows that most individuals derive from castrated rather than male cattle. The largest lowland sites, i.e., the central settlements of Roseldorf and Haselbach show the highest percentage of castrated cattle (67.6% and 48.4%, respectively). When combining the distribution of sexes with age classes, it seems that a high number of adults deriving from later age stages (in several cases between 7 and 10 years) were slaughtered. Later age stages are usually connected to secondary exploitation, suggesting the vital role of cattle as suppliers of milk and labor animals in lowland settlements [79,81,82,85,112]. Taking into consideration the increase in the numbers of oxen and the higher percentage of cattle slaughtered at older age stages, it can be suggested that the demand for working animals increased, pointing towards an intensification of agricultural production.

This observation is further supported by paleopathological finds, which have been connected to working animals (e.g., [82,85,90,92,112]), including finds from Haselbach. Keeping and maintaining cattle and especially oxen is a very demanding task when taking into consideration the low birth rate of cows and the large amount (and good quality) of fodder that castrated animals require. Especially fodder availability is closely related to the availability of grazing land and therefore to the economic situation of the producers/peasants. Such an increase in the numbers of older castrated individuals indicates a well-organized system that could support this change.

There are, of course, exceptions to this general trend: First, the lowland settlement with a quadrangular enclosure of Mšecké Žehrovice (LT C2-D1), where a higher number of younger cattle individuals was noted [89], suggesting an emphasis on meat production. Second, the lowland settlement of Michelndorf [81], where males (41.7%) prevailed over oxen (12.5%), is a unique pattern among the assemblages studied.

In contrast with cattle, sheep/goats were slaughtered more frequently at younger age stages in several lowland settlements (e.g., Giarmata, Inzersdorf, Michelndorf, Oberschauersberg), suggesting meat consumption. However, secondary exploitation of sheep/goat (wool, milk) has also been documented (e.g., Nitra Šindolka, Szakály-Réti Földek, Roseldorf, Haselbach).

Pigs were usually slaughtered between 1.5 and 3 years, which is regarded as an optimal age stage for meat consumption, while the sex distribution might vary (e.g., [81,82,85,105,107]). Pigs are ideal meat suppliers, because they are sexually active at a very young age, and they produce a high number of piglets. Furthermore, they pose less demand for fodder and do not compete with other species; they can find fodder in the woods and even from human kitchen waste. Thus, slaughtering at a young age stage is quite common (e.g., [126,127]).

Oppida

Unfortunately, the oppida of Staré Hradisko and Bratislava in our area of study have not yet provided data on animal sex and age profiles. Looking towards the West, at the oppida of Manching [100] and Altenburg [99] in Southern Germany, cattle was an important meat supplier, indicated by the relatively high number of younger cattle individuals (younger than five years). Although in both oppida females prevail, it is noteworthy that male/castrated cattle reached a comparable high percentage with 45.3% in Altenburg. In Manching sheep/goats deriving from older age stages suggest an emphasis on secondary exploitation (wool, milk), whereas in Altenburg most animals were slaughtered between 2 and 4 years, indicating an emphasis on meat consumption. Most of the pig remains were slaughtered between 2 and 3 years at both sites, proving their role as meat suppliers.

Mining site

The best-studied mining site in the region of interest is the salt mine Bad Dürrnberg [73,74,75,76,77]. The combination of age and sex profiles shows that animals were delivered to the miners. Concerning cattle, the clear selection of adults constitutes an important indicator of this selective strategy. In contrast to the two central lowland settlements (Roseldorf and Haselbach), females prevail among cattle. Concerning sheep/goat, females prevail; although secondary exploitation (wool, milk), represented by some older animals, has been documented, the higher numbers of animals between 5 and 7 years (34.6%) shows that meat acquisition was of primary importance. Finally, the majority of pigs, similarly to other contexts, were slaughtered at the best age stage for meat consumption (44.8%).

Ritual contexts

The examples of Roseldorf-Great Sanctuary [86] and Frauenberg [78] show that the ritual character of these sites is also displayed in the faunal assemblages. The animal bones from Frauenberg reveal a systematic selection of male/castrated animals for all three major domesticated species (more than 80%). Such profiles in the sex distribution support supply of the ritual center from outside. On the other hand, Roseldorf-Great Sanctuary shows higher numbers of females for sheep/goat (78%) and pigs (60%) indicating a kind of flexibility, probably depicting the connection to the economy of the settlement and/or differences based on the worship.

### 5.3. Cattle Make the Difference: Patterns of Husbandry Based on Cattle Remains

According to our study of various archaeozoological aspects from the La Tène sites of Central Europe, it seems that cattle belong to the key species to better understand patterns of economic organization. Variations in the age and sex distribution of cattle proved to be decisive, in contrast to other species that present more flexible biological profiles in the same archaeological contexts. This observation shows the economic importance of cattle for the La Tène Culture, and it can also be used as an indicator of socio-economic status: affording the tender meat of a young cattle (an extremely important animal economically also due to secondary exploitation) or introducing a new cattle morphotype from the South would have been restricted to a (small) part of the human population.

Based on cattle finds, several patterns of cattle husbandry can be discerned among the La Tène period faunal assemblages (cf. [121]). Previous works have already shown the great potential of cattle in better understanding vital aspects of socio-economic organization (e.g., [128,129,130]). In this case, it should be mentioned that our effort of categorizing the various economic systems cannot grasp case studies that have been labeled as “exceptions” (e.g., Michelndorf) or possible intermediate variants that certainly existed. Furthermore, some factors are not yet understood and are of vital importance when discussing economic organization, such as the political system, models of ownership, access to markets, etc. This is especially true when examining so many assemblages from areas demonstrating considerable differences, such as local culture, ecology, and topography. Despite these differences, the following recurring patterns can be distinguished among the available archaeozoological record:

(a) Dominance of females (usually around 70%) together with a wide distribution of different age stages, but with a higher number of animals slaughtered at an older age stage (older than 7 years). This pattern suggests an emphasis on secondary exploitation (milk, labor animals) and meat production. This would be for example the case for the lowland settlement sites of Inzersdorf, Mitterretzbach (ELT), Sajópetri, and Szakály-Réti Földek.

(b) Prevalence of females, but with a higher proportion of male and especially castrated cattle (together around 40–50%) coming from an older age stage (older than 7 years). This pattern would suggest an interest in secondary exploitation, but due to the stronger presence of oxen, it mainly displays the need for more labor animals. This pattern appears for instance in Michelstetten with 47% of male/castrated animals coming from older age stages.

(c) Dominance of adult males/castrated animals (more than 50%), coming from older age stages. This pattern suggests an even higher demand for working animals, either for plowing or for cart transport, which was partially supplied from other settlements, demonstrated by the unnaturally high number of male/castrated cattle. Pattern (c) is characteristic of the lowland central settlements of Roseldorf and Haselbach, which seem to import cattle (probably young oxen) from other lowland settlements (probably from category (a) to use them as draught animals. In the case of Roseldorf, older animals (including cattle) were not only slaughtered for consumption in the settlement but also as part of feastings testified at the sanctuaries (Figure 10).

(d) Prevalence of females showing a clear age selection. This pattern has been noted at the salt mining site of Dürrnberg and it would also indicate supply from outsiders (Figure 10), similar to pattern (c). However, in contrast to the system of supply of the central settlements, the dominance of female cattle in Dürrnberg suggests its important role as a meat supplier after secondary milk exploitation, whereas, in the central settlements of Roseldorf and Haselbach, the importance of cattle mostly lies in its role as labor animals.

(e) Prevalence of female cattle and a higher percentage of younger animals (younger than 7 years). This pattern, typical for the two oppida (Manching and Altenburg) and the quadrangular enclosure of Mšecké Žehrovice examined, suggests an emphasis on the consumption of good quality meat. Although it is not based on statistically significant numbers, this pattern also seems to appear at the site of Wien—“Palais Rasumofsky” [113]—which is part of a very large lowland settlement center.

### 5.4. A Last Aspect: Cynophagy and Hippophagy

One remarkable observation in La Tène sites of Central Europe is related to cynophagy and hippophagy. The documentation of butchery marks on dog and horse bones indicates that the dog and horse meat was (commonly) eaten [79,80,82,91,92,93,94,101,107,113,131]. Consumption of dog and horse meat took place also in the present case study of Haselbach. Furthermore, there is evidence that hippophagy constituted part of ritual activities, as documented at the site of Roseldorf-Great Sanctuary [86] and probably also for a square enclosure at Haselbach [132], whereas dog meat might have been also consumed as part of ritual meals, as the sanctuary of Liptovská Mara would suggest [133].

## 6. Conclusions

The present work is divided into two parts: (a) the archaeozoological analysis of the Middle La Tène settlement in Haselbach, (b) a detailed review of existing archaeozoological data from the La Tène period (c. 450 BC to the end of the first century BC) in Central Europe, from different types of sites, including lowland settlements, hilltop settlements, central settlements, oppida and assemblages of ritual activity.
Concerning the new results from the medium-sized lowland settlement center of Haselbach, the analysis of the faunal remains suggests an economic organization different from smaller categories of lowland settlements. The prevalence of older castrated cattle (partly with pathologies) at Haselbach suggests that the demand for an animal workforce for plowing and transport considerably increased and that cattle from other (rural) settlements were delivered to the site, indicating increasing inter-dependencies and centralization of the settlement network. Additionally, the selection of young pigs further supports the urban character of the site of Haselbach. The data from Haselbach bear striking similarities to the large central settlement in Roseldorf, exhibiting a very similar archaeozoological profile and showing features of proto-urbanism.The observations from Haselbach fit very well with the detailed review of existing archaeozoological data from the Eastern La Tène Culture (55 faunal assemblages from 46 sites in nine countries). These data reveal faunistic changes in the composition, sex, and age profiles of domestic animals that indicate profound socio-economic changes during the Middle La Tène period. The prevalence of older male/castrated cattle in Haselbach is in accordance with profiles gained by other Middle La Tène sites, where a higher percentage of older male/castrated cattle was noted, suggesting a higher demand for labor animals. These gradual changes during the La Tène period could be associated with agricultural intensification, based on the increasing exploitation of labor animals. Finally, our review shows the crucial economic role of cattle for the La Tène period cultures and distinguishes several patterns of cattle husbandry, illustrating that cattle did make the difference.

## Figures and Tables

**Figure 1 animals-13-01847-f001:**
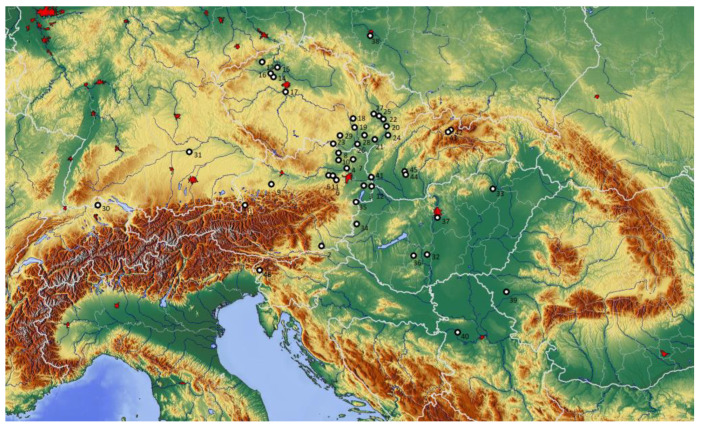
Map of La Tène period faunal assemblages in the Eastern La Tène Culture (the numbers correspond with those in Chapter 4, Pages 12–14).

**Figure 2 animals-13-01847-f002:**
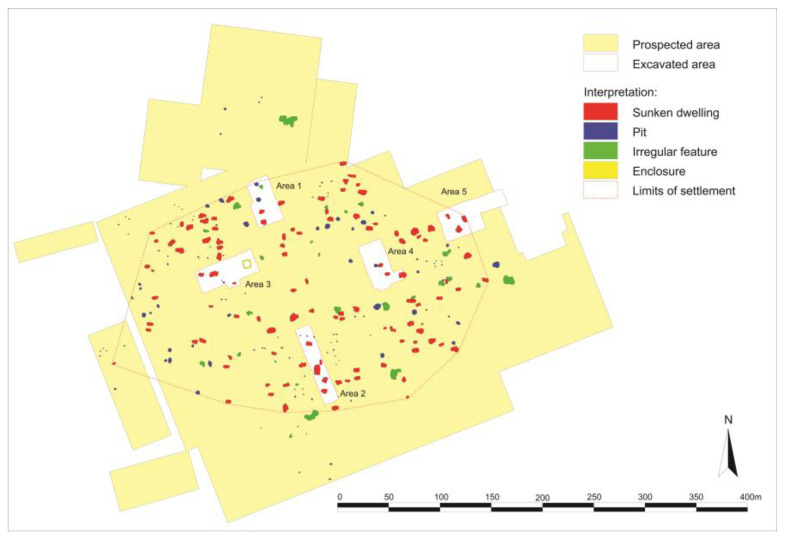
Plan of the Middle La Tène period central lowland settlement of Haselbach in Lower Austria, based on geomagnetic survey and excavations from 2015 to 2019. Graphics: Peter Trebsche.

**Figure 3 animals-13-01847-f003:**
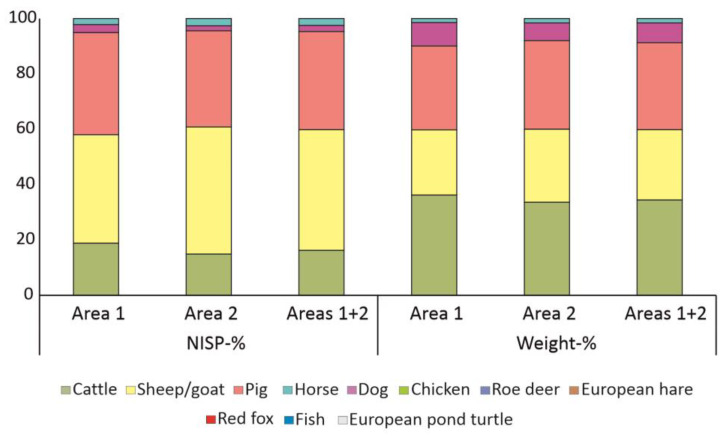
Abundance (NISP % and weight %) of domesticated and wild species from Haselbach. The diagram shows clear differences between NISP % data and weight.

**Figure 4 animals-13-01847-f004:**
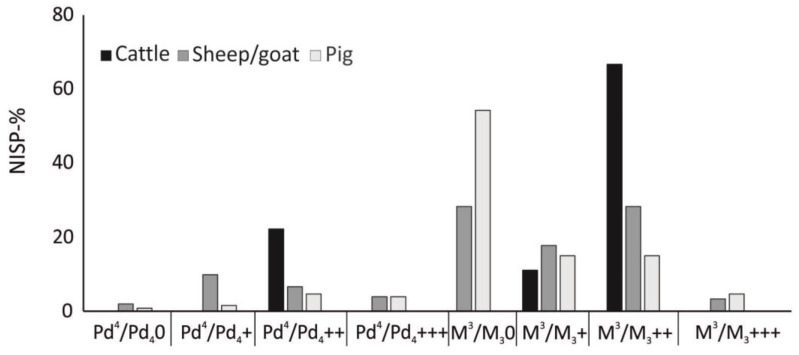
Age reconstruction in Haselbach for the main domesticated animals based on maxillary and mandibular teeth (NISP %). The different age spectrum among the species indicates different economic exploitation. Cattle n: 7 teeth, one cattle cranium and one maxilla fragment, sheep/goat n: 152 teeth, pig n: 127 teeth.

**Figure 5 animals-13-01847-f005:**
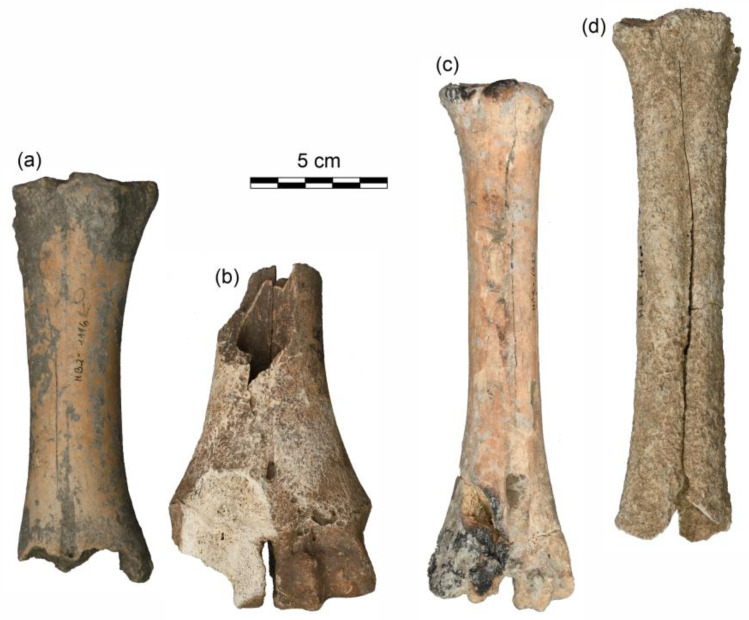
Metacarpi from two male cattle individuals from Haselbach: (**a**) small-sized cattle population (find nr. HB2-1116), (**b**) large-sized cattle population (find nr. HB2-1510); metatarsi from two castrated cattle animals from Haselbach: (**c**) small-sized cattle population (find nr. HB2-1033), (**d**) large-sized cattle population (find nr. HB2-410). Remains of the large-sized cattle population have been found extremely rarely in Iron Age assemblages. The presence of this morphotype in Haselbach (**b**,**d**) is remarkable.

**Figure 6 animals-13-01847-f006:**
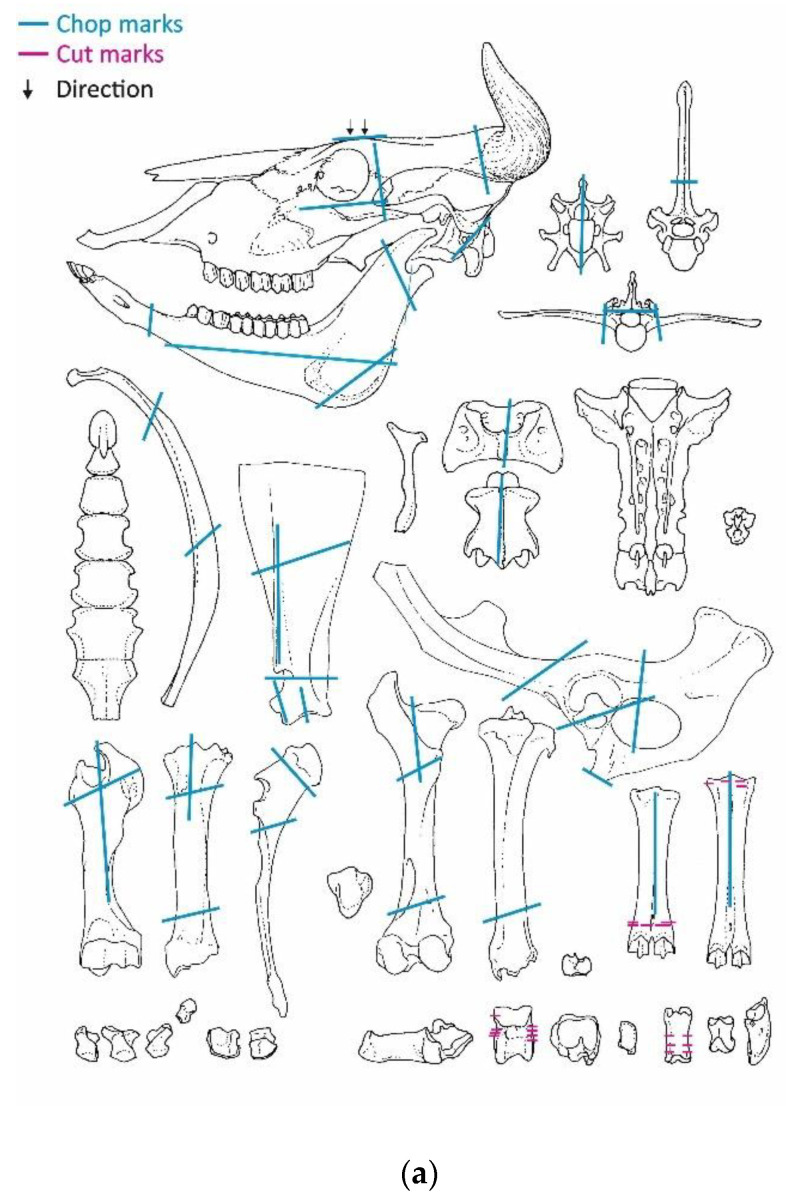
Butchery marks from Haselbach recorded on bones of (**a**) cattle, (**b**) sheep/goat, pig, dog, (**c**) horse.

**Figure 7 animals-13-01847-f007:**
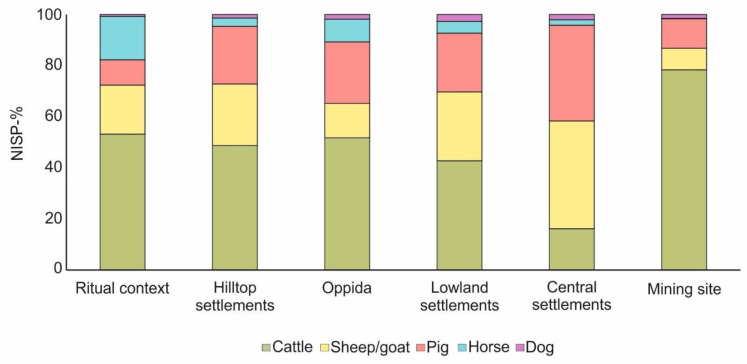
Representation of the most important domesticated species from different types of La Tène sites (see Table 6). Lowland settlements: 38 assemblages, central settlements: 2 assemblages, oppida: 6 assemblages, hilltop settlements: 4 assemblages, ritual contexts: 4 assemblages, mining site: 1 assemblage.

**Figure 8 animals-13-01847-f008:**
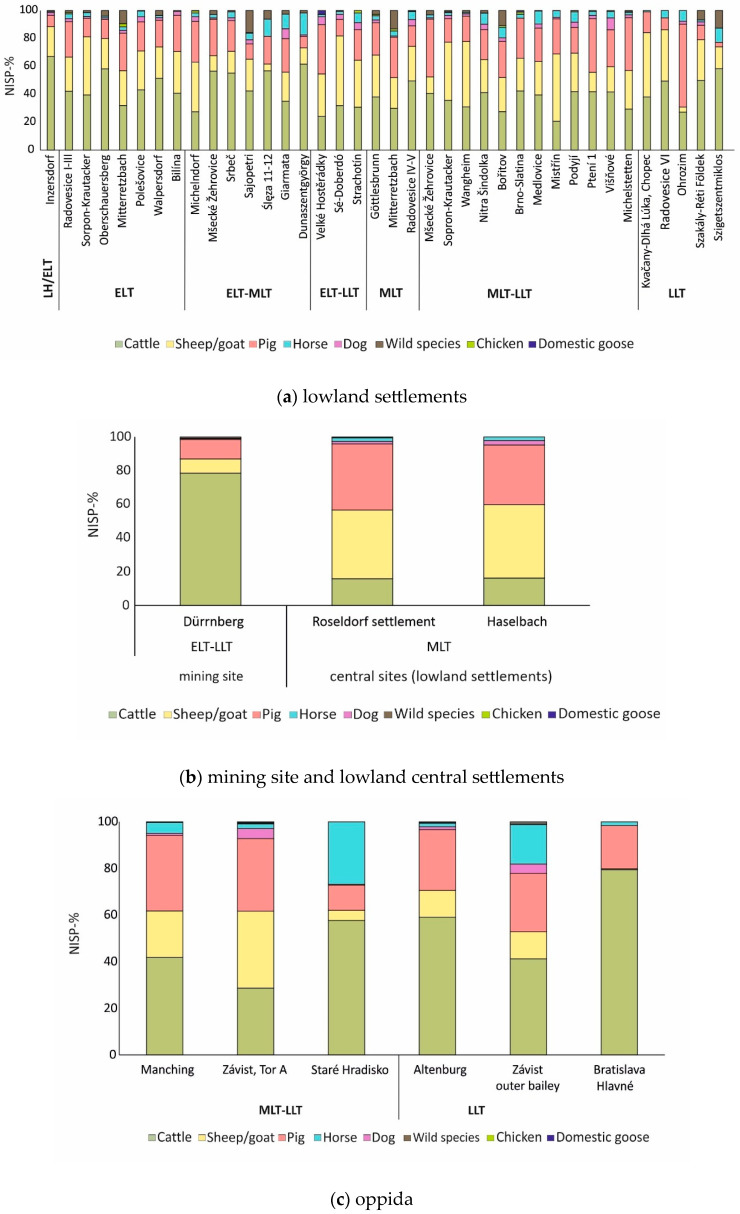
(**a**) Representation of species from La Tène lowland settlements (n: 38) in chronological order. Inzersdorf (NISP: 250), Radovesice I-III (NISP: 2500), Sopron-Krautacker (NISP: 614), Oberschauersberg (NISP: 566), Mitterretzbach (NISP: 1326), Polešovice (NISP: 245), Walpersdorf-Nord (NISP: 1774), Bilína (NISP: 177), Michelndorf (NISP: 1685), Mšecké Žehrovice (NISP: 1416), Srbeč (NISP: 256), Sajópetri (NISP: 2616), Ślęza 11–12 (NISP: ? only percentage), Giarmata (NISP: 393), Dunaszentgyörgy (NISP: 172), Velké Hostěrádky (NISP: 450), Sé-Doberdó (NISP: 988), Strachotín (NISP: 301), Göttlesbrunn (NISP: 1317), Mitterretzbach (NISP: 1274), Radovesice IV-V (NISP: 1173), Mšecké Žehrovice (NISP: 3463), Sopron-Krautacker (NISP: 352) Wangheim (NISP: 340), Nitra Šindolka (NISP: 1349), Bořitov (NISP: 712), Brno-Slatina (NISP: 901), Medlovice (NISP: 987), Mistřín (NISP: 3503), Podyjí (NISP: 1028), Ptení 1 (NISP: 348), Višňové (NISP: 307), Michelstetten (NISP: 2003), Kvačany-Dlhá Lúka (NISP: 213), Radovesice VI (NISP: 95), Ohrozim (NISP: 194), Szakály-Réti Földek (NISP: 449), Szigetszentmiklos (NISP: 127). Abbreviations: LH: Late Hallstatt period, ELT: (**b**) Representation of species from two La Tène lowland central settlements and the mining site of Dürrnberg in chronological order. Dürrnberg (NISP: 15,589), Roseldorf-settlement (NISP: 6569), Haselbach (NISP: 6181). (**c**) Representation of species from La Tène oppida (n: 6) in chronological order. NISP data: Manching (NISP: 388,952), Závist-Gate A (NISP: 6316), Staré Hradisko (NISP: > 10,000), Závist-Outer bailey (NISP: 419), Bratislava-Hlavné námestie 7 (NISP: 296). Abbreviations: MLT: Middle La Tène, LLT: Late La Tène. (**d**) Representation of species from La Tène ritual contexts (n: 4) in chronological order. Kobarid-Bizjakova hiša (NISP: 1219), Roseldorf-Great Sanctuary (NISP: 10,660), Frauenberg (NISP: 8027), Liptovská Mara I (NISP: 1918). Abbreviations ELT: Early La Tène, MLT: Middle La Tène, LLT: Late La Tène. (**e**) Representation of species for the hilltop settlements (n: 4). NISP data: Gomolava VI (NISP: 2988), Liptovská Mara II (NISP: 4708), Nitra-Malý Seminár (NISP: 328), Nitra-Hrad/Východné nádvorie (NISP: 235). Abbreviations: MLT: Middle La Tène, LLT: Late La Tène.

**Figure 9 animals-13-01847-f009:**
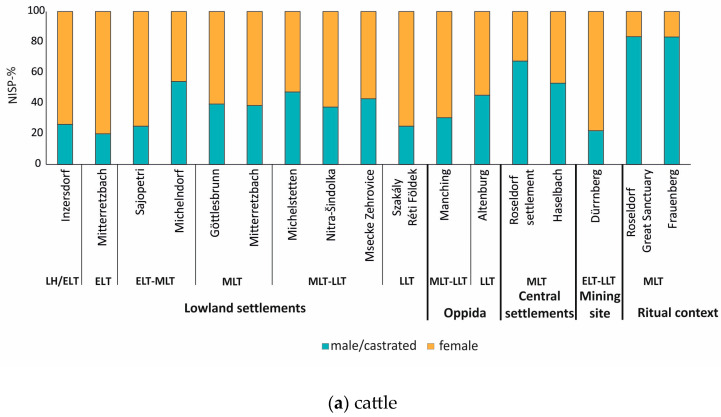
(**a**) Sex estimation for cattle for all archaeological types presented in the current study. NISP data: Inzersdorf (NISP: 61), Mitterretzbach (NISP: 10), Sajópetri (NISP: 16), Michelndorf (NISP: 24), Göttlesbrunn (NISP: 38), Mitterretzbach (NISP: 13), Michelstetten (NISP: 19), Nitra Šindolka (NISP: 16), Mšecké Žehrovice (NISP: 7), Szakály-Réti Földek (NISP only in percentage available), Manching (NISP: 1078), Altenburg (NISP: 457), Roseldorf-settlement (NISP: 34), Haselbach (NISP: 64), Dürrnberg (NISP: 761), Roseldorf-Great Sanctuary (NISP: 394); Frauenberg (NISP: 12). Abbreviations: LH: Late Hallstatt period, ELT: Early La Tène, MLT: Middle La Tène, LLT: Late La Tène. (**b**) Sex estimation for sheep/goats for all site types is summarized in the present study. Inzersdorf (NISP: 7), Michelndorf (NISP: 8), Nitra Šindolka (NISP: 8), Roseldorf-settlement (NISP: 86), Haselbach (NISP: 87), Roseldorf-Great Sanctuary (NISP: 50), Frauenberg (NISP: 4), Manching (NISP: 203), Altenburg (NISP: 154), Dürrnberg (NISP: 36). Abbreviations: LH: Late Hallstatt period, ELT: Early La Tène, MLT: Middle La Tène, LLT: Late La Tène. (**c**) Sex estimation for pig for all site types presented in the current study (based on alveoli, when data were available). Inzersdorf (NISP: 34), Mitterretzbach (NISP: 11), Michelndorf (NISP: 30), Göttlesbrunn (NISP: 10), Mitterretzbach (NISP: 13), Michelstetten (NISP: 60), Nitra Šindolka (NISP: 8), Manching (NISP: 548), Altenburg (NISP: 970), Roseldorf-settlement (NISP: 74), Haselbach (NISP: 27), Roseldorf-Great Sanctuary (NISP: 92), Frauenberg (NISP: 39), Dürrnberg (NISP: 57). Abbreviations: LH: Late Hallstatt period, ELT: Early La Tène, MLT: Middle La Tène, LLT: Late La Tène.

**Figure 10 animals-13-01847-f010:**
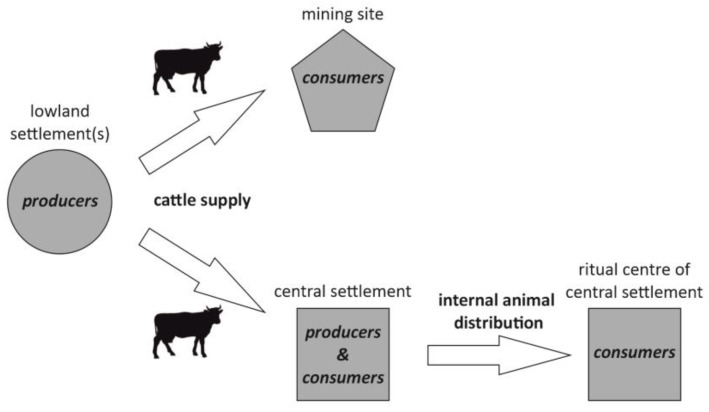
Graphic depiction of cattle supply and distribution based on the biological profiles (age and sex data) reconstructed for the central settlement Roseldorf and its ritual district (Great Sanctuary) and the mining site of Dürrnberg.

**Table 1 animals-13-01847-t001:** Faunal composition from area 1 in Haselbach.

Area 1—NISP							
Element	Cattle	Sheep/Goat	Pig	Horse	Dog	European Hare	European Pond Turtle
Processus frontalis	2	2	-	-	-	-	-
Calva	24	28	54	-	6	-	-
Maxilla	25	90	89	7	8	-	-
Mandibula	48	147	174	8	11	-	-
Vertebrae	36	25	18	1	3	-	-
Costae	58	161	48	-	1	-	-
Scapula	14	22	34	1	2	-	-
Humerus	20	49	53	5	2	1	-
Radius	19	56	24	2	4	-	-
Ulna	9	14	46	-	3	-	-
Carpalia	8	2	-	-	-	-	-
Metacarpalia	18	32	29	1	1	-	-
Pelvis	17	24	23	-	-	-	-
Femur	14	21	33	2	1	-	-
Patella	2	2	-	-	-	-	-
Tibia	23	72	41	3	2	-	-
Fibula/Mall.	1	-	11	2	-	-	-
Talus	3	10	11	2	-	-	-
Calcaneus	9	7	18	-	1	-	-
Tarsalia	3	-	4	3	-	-	-
Metatarsalia	11	32	24	2	-	-	-
Metapodia	5	8	11	6	-	-	-
Phalanx 1	15	10	14	7	1	-	-
Phalanx 2	8	8	11	3	-	-	-
Phalanx 3	7	1	6	5	-	-	-
Sesamoidea	-	-	-	1	-	-	-
Total	399	823	776	61	46	1	1

**Table 2 animals-13-01847-t002:** Faunal composition from area 2 in Haselbach.

Area 2—NISP									
Element	Cattle	Sheep/Goat	Pig	Horse	Dog	Chicken	Roe Deer	Red fox	Fish
Processus frontalis	5	29	-	-	-	-	-	-	-
Calva	41	62	106	-	5	-	-	-	-
Maxilla	21	117	153	3	16	-	-	-	-
Mandibula	54	258	262	10	21	-	-	1	-
Vertebrae	83	45	75	8	6	-	-	-	-
Costae	108	531	98	2	-	-	-	-	-
Coracoid	-	-	-	-	-	1	-	-	-
Scapula	32	56	62	4	6	-	-	-	-
Humerus	33	108	61	-	3	1	-	-	-
Radius	34	117	41	5	8	-	-	-	-
Ulna	8	33	54	-	7	-	-	-	-
Carpalia	10	2	-	-	-	-	-	-	-
Metacarpalia	18	78	70	3	5	-	-	-	-
Pelvis	18	65	50	9	1	-	-	-	-
Femur	23	41	74	5	1	-	-	-	-
Patella	2	-	2	-	-	-	-	-	-
Tibia	27	127	66	2	3	-	1	-	-
Fibula/Mall.	-	-	44	2	-	-	-	-	-
Talus	10	18	15	3	-	-	-	-	-
Calcaneus	12	15	29	3	1	-	-	-	-
Tarsalia	2	5	1	3	-	-	-	-	-
Metatarsalia	16	105	70	3	3	-	-	-	-
Metapodia	13	12	18	2	19	-	-	-	-
Phalanx 1	16	22	29	4	1	-	-	-	-
Phalanx 2	11	16	16	3	-	-	-	-	-
Phalanx 3	9	6	19	-	-	-	-	-	-
Total	606	1868	1415	74	106	2	1	1	1

**Table 3 animals-13-01847-t003:** Sex reconstruction for cattle from areas 1 and 2 in Haselbach.

*Bos*	Male	Male?	Castrated	Castrated?	Female	Female?
Processus frontalis	-	-	2	-	5	-
Pelvis	-	-	4	1	6	-
Metacarpus	2	1	12	1	9	-
Metatarsus	-	-	11	-	6	2
Crania	-	-	-	-	2	-
Total	2	1	29	2	28	2

**Table 4 animals-13-01847-t004:** Sex reconstruction for sheep/goat from areas 1 and 2 in Haselbach.

*Ovis*	Male	Female
Processus frontalis	13	15
Pelvis	22	26
*Total*	*35*	*41*
** *Capra* **	**male**	**female**
Processus frontalis	5	4
Pelvis	-	2
*Total*	*5*	*6*

**Table 5 animals-13-01847-t005:** Sex reconstruction for pig from areas 1 and 2 in Haselbach (* not in alveoli).

*Sus*		Male	Female
Maxilla	Alveoli	1	6
	Canini *	30	24
Mandibula	Alveoli	12	8
	Canini *	53	34
Total		96	72

**Table 6 animals-13-01847-t006:** Detailed information on the archaeological sites used for the archaeozoological analysis presented in alphabetical order and per modern country (see Figure 1 for the location of the sites on the map). Abbreviations: Ha: Hallstatt period, LT: La Tène, ELT: Early La Tène, MLT: Middle La Tène, LLT: Late La Tène.

Nr.	Site/Assemblage	Country	Dating	Exact Dating	Type	Archaeozoological Literature
**1.**	Dürrnberg	Austria	ELT-LLT		Salt mine	[73,74,75,76,77]
**2.**	Frauenberg near Leibnitz	Austria	M-LLT	LT C2-D	Sanctuary	[78]
**3.**	Göttlesbrunn	Austria	MLT	LT C1-2	Lowland settlement	[79]
**4.**	Haselbach (Area 1–2)	Austria	MLT	LT B2-LT D1, main phase LT C	Lowland central settlement	present work
**5.**	Inzersdorf	Austria	ELT	Ha D3/LT A to LT C	Lowland settlement	[80]
**6.**	Michelndorf	Austria	ELT-MLT	LT B2-C2	Lowland settlement	[81]
**7.**	Michelstetten	Austria	M-LLT	LT C1-D1	Lowland settlement	[82]
**8.**	Mitterretzbach	Austria	ELT	-	Lowland settlement	[83]
			MLT			
**9.**	Oberschauersberg	Austria	ELT	LT A	Lowland settlement	[84]
**10.**	a. Roseldorf-settlement	Austria	MLT	-	Lowland central settlement	[85]
	b. Roseldorf-Great Sanctuary (Object 1)			-	Sanctuary	[86]
**11.**	Walpersdorf-Nord	Austria	ELT	-	Lowland settlement	[80]
**12.**	Wangheim	Austria	M-LLT	-	Lowland settlement	[87]
**13.**	Bilína	Czech Republic (Bohemia)	ELT	LT B1a	Lowland settlement	[88]
**14.**	Mšecké Žehrovice	Czech Republic (Bohemia)	E-MLT	LT B2-C1	Lowland settlement	[89]
			M-LLT	LT C2-D1	Lowland settlement (quadrangular enclosure)	
**15.**	a. Radovesice I-III	Czech Republic (Bohemia)	ELT	Ha D-LT B1	Lowland settlement	[90]
	b. Radovesice IV-V		MLT	LTB2-LTC/D		
	c. Radovesice VI		LLT	LT D		
**16.**	Srbeč	Czech Republic (Bohemia)	E-MLT	LT B2-C1	Lowland settlement	[91]
**17.**	a. Závist, Gate A, Hor. II-V	Czech Republic (Bohemia)	M-LLT	LT C2-D2	Oppidum	[92]
	b. Závist-Outer bailey		LLT	LT D1	Oppidum (outer bailey)	[93]
**18.**	Bořitov	Czech Republic (Moravia)	M-LLT	LT C2-D1/D2	Lowland settlement	[94]
**19.**	Brno-Slatina	Czech Republic (Moravia)	M-LLT	LT C-D1	Lowland settlement	[70]
**20.**	Medlovice	Czech Republic (Moravia)	-	-	Lowland settlement	[70]
**21.**	Mistřín	Czech Republic (Moravia)	-	-	Lowland settlement	[70]
**22.**	Ohrozim	Czech Republic (Moravia)	LLT	-	Lowland settlement	[70]
**23.**	Podyjí	Czech Republic (Moravia)	-	-	Lowland settlement	[70]
**24.**	Polešovice	Czech Republic (Moravia)	ELT	-	Lowland settlement	[95]
**25.**	Ptení 1	Czech Republic (Moravia)	-	-	Lowland settlement	[70]
**26.**	Strachotín	Czech Republic (Moravia)	-	-	Lowland settlement	[96]
**27.**	Staré Hradisko	Czech Republic (Moravia)	M-LLT	LT C2-D	Oppidum	[97]
**28.**	Velké Hostěrádky	Czech Republic (Moravia)	E-LLT	LTB-D	Lowland settlement	[98]
**29.**	Višňové	Czech Republic (Moravia)	-	-	Lowland settlement	[70]
**30.**	Altenburg-Rheinau	Germany	LLT		Oppidum	[99]
**31.**	Manching	Germany	M-LLT	LTC-D1	Oppidum	[100]
**32.**	Dunaszentgyörgy	Hungary	E-MLT	LT B2-C1	Lowland settlement	[101]
**33.**	Sajópetri	Hungary	E-MLT	LT B2-C1	Lowland settlement	[102]
**34.**	Sé-Doberdó	Hungary	E-LLT	Ha D-LT D	Lowland settlement	[103]
**35.**	Sopron-Krautacker	Hungary	ELT	Ha D-LT A	Lowland settlement	[104]
			M-LLT	LT B-LT D		
**36.**	Szakály-Réti Földek	Hungary	LT	-	Lowland settlement	[105]
**37.**	Szigetszentmiklos	Hungary	LLT	LT D	Lowland settlement	[106]
**38.**	Ślęza 11–12	Poland	E-MLT	LT B2-C1	Lowland settlement	[72]
**39.**	Giarmata	Romania	E-MLT	LT B2-C	Lowland settlement	[107]
**40.**	Gomolava VI	Serbia	M-LLT	LT C2-D	Hilltop settlement	[108]
**41.**	Bratislava-Hlavné námestie 7	Slovakia	LLT	LT D2	Oppidum-(outer bailey)	[71]
**42.**	Kvačany-Dlhá Lúka	Slovakia	M-LLT	LT C-D	Lowland settlement	[71]
**43.**	a. Liptovská Mara I	Slovakia	LLT	LT D1-D2	Sanctuary	[71,109]
	b. Liptovská Mara II	Slovakia	M-LLT	(Ha D1-LT B to C2) main phase: LT C2	Hilltop settlement	[71,109]
**44.**	a. Nitra-Hrad/Východné nádvorie	Slovakia	LLT	LT D	Hilltop settlement	[71]
	b. Nitra-Malý Seminár	Slovakia	LLT	LT D	Hilltop settlement	[71,110]
**45.**	Nitra Šindolka	Slovakia	M-LLT	LTC2-D1	Lowland settlement	[71,111]
**46.**	Kobarid-Bizjakova hiša	Slovenia	E-MLT	LT B2-C	Ritual site	[112]

**Table 7 animals-13-01847-t007:** Estimation of height at withers for cattle from selected La Tène sites and Haselbach. Abbreviations: Mp: metapodials, Mc: metacarpals, Mt: metatarsals.

Cattle—Height at Withers (cm)					
Site	Min	Max	Element	N	References
Dürrnberg-Ramsautal (female)	94.8	111.1	Mp	58	[73]
Dürrnberg-Ramsautal (castrated)	103.2	122.5	Mp	33	[73]
Gomolava	101.0	125.0	Mc	9	[108]
Gomolava	92.0	116.5	Mt	9	[108]
Haselbach (female)	106.7	111.0	Mc	2	present study
Haselbach (castrated)	98.3	113.1	Mp	5	present study
Inzersdorf-Walpersdorf (female)	104.0	107.1	Mp	4	[80]
Mšecké Žehrovice (LTC2-D1)	103.8	112.5	Mc	5	[89]
Mšecké Žehrovice (LTC2-D1)	99.6	103.4	Mt	2	[89]
Roseldorf-settlement (castrated)	106.6	116.2	Mp	7	[85]
Roseldorf-Great Sanctuary (female)	105.8	108.2	Mp	5	[86]
Roseldorf-Great Sanctuary (castrated)	100.1	117.1	Mp	27	[86]

**Table 8 animals-13-01847-t008:** Metric comparison (mm) of various cattle bones from selected La Tène sites and Haselbach, based on Driesch (1976). Abbreviations: SLC: smallest length of the collum scapulae, Bp: (greatest) breadth of the proximal end, Bd: (greatest) breadth of the distal end, GLl: greatest length of the lateral half, Glpe: greatest length of the peripheral half.

Element/Site	Measurement	Min	Max	Average	N	References
**Scapula**	**SLC**					
Dürrnberg-Ramsautal		34.0	54.0	42.6	81	[73]
Göttlesbrunn		40.0	53.5	46.6	9	[79]
Haselbach		33.0	51.5	45.4	8	present study
Roseldorf-Great Sanctuary		41.5	57.5	49.7	32	[86]
**Radius**	**Bp**					
Dürrnberg-Ramsautal		61.5	83.0	69.0	101	[73]
Haselbach		54.0	78.5	70.1	9	present study
Roseldorf-Great Sanctuary		68.5	86.5	77.7	43	[86]
**Metacarpus**	**Bp**					
Dürrnberg Ramsautal		43.5	59.5	48.8	68	[73]
Haselbach		38.5	60.5	50.3	17	present study
Roseldorf-Great Sanctuary (female)		45.5	52.0	48.3	6	[86]
Roseldorf-Great Sanctuary (castrated)		48.5	61.5	55.7	47	[86]
**Tibia**	**Bd**					
Dürrnberg-Ramsautal		45.5	63.5	53.2	175	[73]
Göttlesbrunn		50.5	62.0	56.2	6	[79]
Haselbach		48.5	60.0	55.0	12	present study
Roseldorf-Great Sanctuary		51.5	65.0	58.1	80	[86]
**Talus**	**GLl**					
Dürrnberg-Ramsautal		48.0	64.5	57.4	191	[73]
Göttlesbrunn		46.0	61.5	53.5	26	[79]
Inzersdorf-Walpersdorf		54.5	63.5	60.0	11	[80]
Haselbach		54.0	58.0	56.1	8	present study
Roseldorf-Great Sanctuary		55.0	67.0	60.3	60	[86]
**Metatarsus**	**Bp**					
Dürrnberg-Ramsautal		35.0	50.5	40.8	78	[73]
Haselbach		37.5	49.5	44.4	10	present study
Roseldorf-Great Sanctuary (female)		37.5	44.5	40.9	13	[86]
Roseldorf-Great Sanctuary (castrated)		41.5	48.0	44.5	38	[86]
**Phalanx 1**	**Glpe**					
Dürrnberg-Ramsautal		43.0	58.5	52.0	185	[73]
Inzersdorf-Walpersdorf		50.0	58.5	53.3	12	[80]
Haselbach		48.5	58.0	52.8	20	present study
Roseldorf-Great Sanctuary		47.0	59.0	54.0	39	[86]

**Table 9 animals-13-01847-t009:** Estimation of height at withers for sheep from selected La Tène sites and Haselbach.

Sheep–Height at Withers (cm)				
Site	N	Average	Range of Variation	References
Dürrnberg-Ramsautal	25	65.6	57.2–77.1	[73]
Göttlesbrunn	7	61.0	56.1–65.0	[79]
Inzersdorf-Walpersdorf	9	59.9	56.9–64.1	[80]
Haselbach	41	61.9	52.1–68.7	present study
Michelstetten	9	62.5	57.2–67.3	[82]
Roseldorf-Great Sanctuary	9	63.9	61.2–67.2	[86]
Roseldorf-settlement	19	61.3	59.8–63.1	[85]

**Table 10 animals-13-01847-t010:** Range of variation for the Bd (greatest breadth of the distal end), von den Driesch (1976) of sheep humerus from selected La Tène sites and Haselbach.

Sheep–Humerus Bd (mm)					
Site	N	Min	Max	Average	References
Dürrnberg-Ramsautal	36	27.5	39.0	30.5	[73]
Göttlesbrunn	5	28.5	31.0	29.8	[79]
Inzersdorf-Walpersdorf	8	26.5	32.0	28.9	[80]
Haselbach	46	27.0	35.0	30.5	present study
Nitra Šindolka	6	25.8	31.2	28.2	[111]
Roseldorf-settlement	17	27.0	33.5	30.1	[85]
Roseldorf-Great Sanctuary	12	28.5	33.0	30.3	[86]

**Table 11 animals-13-01847-t011:** Estimation of height at withers for pig from selected La Tène sites and Haselbach.

Pig—Height at Withers (cm)				
Site	N	Average	Range of Variation	References
Dürrnberg-Ramsautal	15	75.0	67.0–81.0	[73]
Göttlesbrunn	8	77.2	69.8–83.6	[79]
Inzersdorf-Walpersdorf	4	78.0	71.0–82.0	[80]
Haselbach	11	70.9	65.1–78.4	present study
Michelstetten	8	76.8	74.0–80.6	[82]
Roseldorf-Great Sanctuary	19	72.8	60.9–82.0	[86]
Roseldorf-settlement	17	74.4	70.3–78.0	[85]

**Table 12 animals-13-01847-t012:** Range of variation for the Bd (greatest breadth of the distal end) of pig tibia from selected La Tène sites and Haselbach, based on von den Driesch (1976).

Pig—Tibia Bd (mm)					
Site	N	Min	Max	Average	References
Dürrnberg-Ramsautal	46	24.5	30.5	27.8	[73]
Gomolava	4	25.0	27.0	26.4	[108]
Göttlesbrunn	5	25.0	30.0	28.0	[79]
Inzersdorf-Walpersdorf	5	26.5	30.0	28.4	[80]
Haselbach	24	25.5	31.5	28.7	present study
Mšecké Žehrovice (LTC2-D1)	5	26.0	29.5	27.4	[89]
Roseldorf-settlement	42	26.5	30.5	28.3	[85]

**Table 13 animals-13-01847-t013:** Length of the cheektooth row (M3–P1) of dog remains from selected La Tène sites and Haselbach, based on von den Driesch (1976).

Dog: Length of the Cheektooth Row (M_3_–P_1_) (mm)					
Site	Min	Max	X	N	References
Dürrnberg-Ramsautal	67.5	80.5	75.3	6	[73]
Gomolava	59.0	77.0	65.9	14	[108]
Haselbach	68.5	79.5	75.1	14	present study
Manching	54.0	80.0	69.4	66	[100]
Michelstetten	68.0	82.0	73.2	3	[82]
Nitra Šindolka	68.3	82.3	73.8	11	[111]
Roseldorf-Great Sanctuary	57.5	76.5	69.5	10	[86]
Roseldorf-settlement	55.4	81.0	72.2	4	[85]

## Data Availability

Data available on request.

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
