# Peer review of "Cattle Make the Difference: Variations and Developments of Animal Husbandry in the Central European La Tène Culture"

_animals, 2023, doi:10.3390/ani13111847_

Round 1

Reviewer 1 Report

This is an excellent contribution, which is distinguished on a methodological level by the depth and quality of the archaeozoological analysis always confronted with historical and archaeological data. The detail of the animal economy of the Haselbach site is projected against the backdrop of an impressive number of sites of the same age in a very broad geographical setting coinciding with the extension of the La Tène culture in middle and south-eastern Europe. The article produces a new socio-economic perspective and is therefore innovative and inspiring for further study. I think in particular that the data collected here could be appropriately compared with the data available for northern Italy. This area, in fact, can be considered fundamental in the transmission of large southern cattle breeds, and its cultural and 'commercial' relations with the areas north and east of the Alpine watershed are well known in the literature (archaeogenetics, isotopic surveys, etc.). 

Author Response

We are grateful to both reviewers for their comments and helpful insights!

 Reviewer 1

 This is an excellent contribution, which is distinguished on a methodological level by the depth and quality of the archaeozoological analysis always confronted with historical and archaeological data. The detail of the animal economy of the Haselbach site is projected against the backdrop of an impressive number of sites of the same age in a very broad geographical setting coinciding with the extension of the La Tène culture in middle and south-eastern Europe. The article produces a new socio-economic perspective and is therefore innovative and inspiring for further study. I think in particular that the data collected here could be appropriately compared with the data available for northern Italy. This area, in fact, can be considered fundamental in the transmission of large southern cattle breeds, and its cultural and 'commercial' relations with the areas north and east of the Alpine watershed are well known in the literature (archaeogenetics, isotopic surveys, etc.). 

Reply: Yes, this is true! We included the overview of Trentacoste et al. 2018.

Reviewer 2 Report

General comments

This is a useful and original article, pulling together known zooarchaeological data for Iron Age sites in a given region to set the findings from one site in context, and make some observations about supply networks.

Overall it is a well-written piece of work, and some minor comments are provided on the manuscript and below, but I do have some major points that should be taken into consideration:

1.       I think you are missing a major point which is that there are a lot more sheep and pigs at your central sites! In an economy where cattle are esteemed as you suggest, what could this mean?? I think that you could make an argument that sheep make the difference at these sites! especially as you make the point that they are likely importing some large breeding stock from far afield, so are sites of some affluence.

2.       The increase in the proportion of cattle at most sites from the MLT is a fundamental point, and when combined with a general increase in males suggests that arable production is becoming more important. The ELT seems more likely to be heavily biased towards smaller, self-sufficient sites, where male cattle are presumably culled early for meat as females are easier to handle and more useful (breeding, milk and traction). This point is nearly made, but you could be more explicit.

3.       The consistency in sheep husbandry over time and at different site types is interesting and deserves mentioning that they would have been vital for wool (as you have males and females in similar quantities).

4.       The major problem I have is your insistence that male cattle dominate the haselbach assemblage, when in fact they are recorded in similar quantities to females and probably reflect an economy with no preferential culling of either sex. I do agree that there are more males at the central sites, and the outlying farms could be sending their old working animals to central sites, but I don’t think it suggests they are importing males, when it could just as easily represent a single herd kept by those living at the central site itself.

Minor points:

It is confusing as you seem to use the terms ‘sheep/ goat’ and ‘small ruminant’ interchangeably, but as the latter could include deer it would be better and less confusing just to use the term ‘sheep/ goat’ instead.

There is no need to use a tenth of a percentile so keep to whole numbers e.g. round 28.3% to 28%.

In the UK the standard term for describing deciduous premolars is to use the convention dP, though I notice you use Pd – is this an Austrian convention? in which case ignore my comments, I just want to check it isn’t a translation mistake.

You don’t need to list all the butchery bone by bone in the text as you present it well enough in the figures. Limit the text to summarising the effect of the butchery as you already have done e.g. horn removal, disarticulation, jointing, marrow and meat removal.

It would be nice to insert a figure after fig 7 to show overall changes in sp rep through time as well as by site type… especially as there appears to be an increase in cattle at many sites from the MLT.

The comparative figures 8a-e are not very easy to read as you have all taxa, including ‘minor’ species in one chart. It would be easier to compare if you concentrated on the proportions of c, s/g and p, especially as you have so few minor taxa from Haselbach.

When describing findings from other sites you don’t need to keep including the references as they are already in Table 6.

My main concern is the emphasis on a ‘dominance’ of male cattle at Haselbach, which I disagree with as the proportion is only 53/47, which is within the natural range. It is possible that males are preferentially sent to these central sites, but it is not as meaningful as you suggest. This bias is illustrated when you say “almost equal distribution of males and females” for pigs (line 652), when the ratio is 96:72 (57% male)! Be consistent and don’t let a good story get in the way of the facts!

In your discussion lines 707-8 the description of lowland settlements could also be one of self-sufficient sites rather than producers.

I have made a few adjustments and have one query on the convention for describing deciduous teeth, but it is generally well written.

Author Response

We are grateful to the editors and both reviewers for their comments and helpful insights!

 Editors comment: Personally, I consider the term small ruminants to be the same as sheep/goat, but please refer to this comment.

Reply: I completely agree with you, but we could change it.

Reviewer 2

General comments

This is a useful and original article, pulling together known zooarchaeological data for Iron Age sites in a given region to set the findings from one site in context, and make some observations about supply networks.

Overall it is a well-written piece of work, and some minor comments are provided on the manuscript and below, but I do have some major points that should be taken into consideration:

  1. I think you are missing a major point which is that there are a lot more sheep and pigs at your central sites! In an economy where cattle are esteemed as you suggest, what could this mean?? I think that you could make an argument that sheep make the difference at these sites! especially as you make the point that they are likely importing some large breeding stock from far afield, so are sites of some affluence.

Reply

“I think you are missing a major point which is that there are a lot more sheep and pigs at your central sites!”

We comment and discuss this point in following lines: 387-392, 660-686.

“In an economy where cattle are esteemed as you suggest, what could this mean?? I think that you could make an argument that sheep make the difference at these sites!”

Actually, in these two sites sheep prevail only numerically, when somebody checks for example the weight analysis, then cattle individuals prevail, suggesting that the meat supply was based on cattle and not on sheep (please see Figure 3 of the paper). If a centralization was taking place in these sites, sheep are easier to keep because they do not need so much space like cattle and they are easier to keep. The reason why we say that cattle make the difference is because changes in the age and sex profiles of cattle from all studied sites (including the central sites) was the key to understand changes in the economy. Sheep profiles might change significantly from site to site. The fact that the number of sheep increases (in some sites incl. the central sites) is also related to changes in the profiles of cattle.

especially as you make the point that they are likely importing some large breeding stock from far afield, so are sites of some affluence.

They are very important sites indeed, as it has been mentioned in lines 340-341. But the importance of these sites did not become visible because of the numerical prevalence of sheep. The differences in the age and sex of cattle were the key to understand the significant character of the central sites.

  1. The increase in the proportion of cattle at most sites from the MLT is a fundamental point, and when combined with a general increase in males suggests that arable production is becoming more important. The ELT seems more likely to be heavily biased towards smaller, self-sufficient sites, where male cattle are presumably culled early for meat as females are easier to handle and more useful (breeding, milk and traction). This point is nearly made, but you could be more explicit.

We have devoted a whole chapter in the discussion: “Review of archaeozoological data from La Tè ne sites in Central Europe”, where we discuss this observation based on the results and we even propose models of economy (actually, this is why we also chose this specific title).  The discussion is the only place where we could discuss it in detail, since this is not a result, but our interpretation from the results we got. At the section of the “results” we just give the data pure, without mixing interpretations.

  1. The consistency in sheep husbandry over time and at different site types is interesting and deserves mentioning that they would have been vital for wool (as you have males and females in similar quantities).

We cannot state this, because wool is not the only reason that sheep was kept. In several sites, sheep has been slaughtered at a younger age stage, suggesting meat supply and not wool exploitation.

  1. The major problem I have is your insistence that male cattle dominate the haselbach assemblage, when in fact they are recorded in similar quantities to females and probably reflect an economy with no preferential culling of either sex. I do agree that there are more males at the central sites, and the outlying farms could be sending their old working animals to central sites, but I don’t think it suggests they are importing males, when it could just as easily represent a single herd kept by those living at the central site itself.

Males/castrated are represented in Haselbach with 53.1%, which means that they are more than the females (and we try to formulate it carefully, very often simply stating “a higher number”). More than half belonged to males/castrated. We do not think that this percentage should be ignored, because it does show a preference and a completely different economic organization. If you take in mind, that the birth rate of cattle is almost 50% males and 50% females and that this percentage is regulated by the peasants, then we do have a higher number of males/castrated. This is a result that we cannot changes and not an interpretation. Please note that this interesting result comes from two very important sites and not from two simple rural sites. In Roseldorf coins were even produced! So, we do not really think that this is a coincidence.

Minor points:

It is confusing as you seem to use the terms ‘sheep/ goat’ and ‘small ruminant’ interchangeably, but as the latter could include deer it would be better and less confusing just to use the term ‘sheep/ goat’ instead.

Reply: Done!

There is no need to use a tenth of a percentile so keep to whole numbers e.g. round 28.3% to 28%.

Reply: usually this info is published like this, so we use the results as published by the authors.

In the UK the standard term for describing deciduous premolars is to use the convention dP, though I notice you use Pd – is this an Austrian convention? in which case ignore my comments, I just want to check it isn’t a translation mistake.

Reply: yes, exactly.

You don’t need to list all the butchery bone by bone in the text as you present it well enough in the figures. Limit the text to summarising the effect of the butchery as you already have done e.g. horn removal, disarticulation, jointing, marrow and meat removal.

Reply: the depiction of the marks could be viewed as a form of interpretation, so we give all data for people that might have a special interest in butchery techniques and methods.

It would be nice to insert a figure after fig 7 to show overall changes in sp rep through time as well as by site type… especially as there appears to be an increase in cattle at many sites from the MLT.

Reply: this is what figures 8a-e show.

The comparative figures 8a-e are not very easy to read as you have all taxa, including ‘minor’ species in one chart. It would be easier to compare if you concentrated on the proportions of c, s/g and p, especially as you have so few minor taxa from Haselbach.

Reply: there are important changes at the profiles of other species too, like the horse shows or the case of the wild species. In order to make it easier, we chose not to show the wild species, but to out them all together as “wild species”.

When describing findings from other sites you don’t need to keep including the references as they are already in Table 6.

Reply: True!

My main concern is the emphasis on a ‘dominance’ of male cattle at Haselbach, which I disagree with as the proportion is only 53/47, which is within the natural range. It is possible that males are preferentially sent to these central sites, but it is not as meaningful as you suggest. This bias is illustrated when you say “almost equal distribution of males and females” for pigs (line 652), when the ratio is 96:72 (57% male)! Be consistent and don’t let a good story get in the way of the facts!

Reply: We have replied already to this comment. If it would not have been meaningful, then we have found it in other sites too. Comparing pigs with cattle is not really possible, since the exploitation is different! The dominance of male pigs happens very often, but not of cattle!

In your discussion lines 707-8 the description of lowland settlements could also be one of self-sufficient sites rather than producers.

Reply: as we mention in lines 711-713 the lowland settlements suggest different economic models…including self-sufficent sites and sites of producers.

Some final remarks

We also thank the reviewer that his/her comments done on the pdf were also summarized separately and so we could give a proper reply.

Comments on some tables: they are not always homogenous (especially the one concerning the metric data) but this is the result of what it is published, or where we had access to or what we could find.

Concerning some comments on the figures done by reviewer 2, of course some things can change, such as orientation or place if the editor has no problem with it!

We can send the manuscript for a language control, if we have the time to do so.